# Inhibition of Calcium/Calmodulin-Dependent Protein Kinase Kinase β Is Detrimental in Hypoxia–Ischemia Neonatal Brain Injury

**DOI:** 10.3390/ijms20092063

**Published:** 2019-04-26

**Authors:** Jia-Wei Min, Fan Bu, Li Qi, Yashasvee Munshi, Gab Seok Kim, Sean P. Marrelli, Louise D. McCullough, Jun Li

**Affiliations:** Department of Neurology, University of Texas Health Science Center, McGovern Medical School, MSER338, 6431 Fannin St, Houston, TX 77030, USA; Jia-Wei.Min@uth.tmc.edu (J.-W.M.); Fan.Bu@uth.tmc.edu (F.B.); Li.Qi@uth.tmc.edu (L.Q.); Yashasvee.Munshi@uth.tmc.edu (Y.M.); Gab.Kim@uth.tmc.edu (G.S.K.); Sean.P.Marrelli@uth.tmc.edu (S.P.M.); Louise.D.McCullough@uth.tmc.edu (L.D.M.)

**Keywords:** hypoxia–ischemia, CaMKK β, neonatal, blood–brain barrier

## Abstract

Neonatal hypoxia–ischemia (HI) is a major cause of death and disability in neonates. HI leads to a dramatic rise in intracellular calcium levels, which was originally thought to be detrimental to the brain. However, it has been increasingly recognized that this calcium signaling may also play an important protective role after injury by triggering endogenous neuroprotective pathways. Calcium/calmodulin-dependent protein kinase kinase β (CaMKK β) is a major kinase activated by elevated levels of intracellular calcium. Here we evaluated the functional role of CaMKK β in neonatal mice after HI in both acute and chronic survival experiments. Postnatal day ten wild-type (WT) and CaMKK β knockout (KO) mouse male pups were subjected to unilateral carotid artery ligation, followed by 40 min of hypoxia (10% O_2_ in N_2_). STO-609, a CaMKK inhibitor, was administered intraperitoneally to WT mice at 5 minutes after HI. TTC (2,3,5-triphenyltetrazolium chloride monohydrate) staining was used to assess infarct volume 24 h after HI. CaMKK β KO mice had larger infarct volume than WT mice and STO-609 increased the infarct volume in WT mice after HI. In chronic survival experiments, WT mice treated with STO-609 showed increased tissue loss in the ipsilateral hemisphere three weeks after HI. Furthermore, when compared with vehicle-treated mice, they showed poorer functional recovery during the three week survival period, as measured by the wire hang test and corner test. Loss of blood–brain barrier proteins, a reduction in survival protein (Bcl-2), and an increase in pro-apoptotic protein Bax were also seen after HI with CaMKK β inhibition. In conclusion, inhibition of CaMKK β exacerbated neonatal hypoxia–ischemia injury in mice. Our data suggests that enhancing CaMKK signaling could be a potential target for the treatment of hypoxic–ischemic brain injury.

## 1. Introduction

Neonatal hypoxia–ischemia (HI) is a leading cause of death and cognitive impairment in children, occurring in approximately 1–3 newborns per 1000 live births [1,2]. During hypoxia–ischemia, anaerobic glycolysis is rapidly initiated due to the absence of oxygen reaching the brain, resulting in an inadequate supply of energy [3]. Following energy failure, extracellular glutamate levels rise due to impaired reuptake and uncontrolled release of glutamate. Subsequently, there is an increase in intracellular free calcium via the over-activation of the N-methyl-D-aspartate receptor and release of calcium from intracellular calcium storage [4]. Intracellular free calcium overload has traditionally been believed to play detrimental roles in the pathology of brain ischemia [5]. However, clinical trials testing calcium blockers failed to improve outcomes in ischemic stroke, possibly due to the nonspecific blockage of calcium-activated pathways [6]. Interestingly, studies have shown that calcium-activated calcium/calmodulin-dependent protein kinase kinase (CaMKK) can participate in specific signal transduction pathways, leading to neuronal survival in pathological conditions [7].

CaMKK, a serine/threonine-specific protein kinase, has two isoforms, α and β, both of which are highly expressed in the brain [8]. Upon activation, CaMKK phosphorylates its two main downstream substrates, CaMK I and IV [9]. Together, they are termed CaMK Cascade. It has been shown that CaMK IV, downstream of CaMKK, inhibits the function of histone deacetylase 4 [10]. Histone deacetylase (HDAC) inhibitors are well known to be neuroprotective in stroke models because they enhance the expression of anti-apoptotic proteins such as B-cell lymphoma 2 (Bcl2) [11]. Additionally, CaMKK signaling may enhance blood–brain barrier (BBB) integrity, the impairment of which is a major pathological component of cerebral ischemia [12,13]. AMP-activated protein kinase (AMPK), a sensor of cellular stress, may also activate CaMKK during the depletion of ATP or alterations in intracellular calcium concentrations [14,15]. It has been shown that the inhibition of AMPK prior to HI results in a worse outcome in mouse models of HI [16]. However, the role of CaMKK in the neonatal brain after a hypoxic–ischemic insult is still not understood. In the present study, we explored the functional role of CaMKK and its downstream mediators in neonatal hypoxia–ischemia brain injury.

## 2. Results

### 2.1. Deletion of CaMKK β Increased Infarct Volume in Neonatal Mice Assessed 24 h after HI

We compared the infarction sizes of neonatal CaMKK β knockout (KO) mice and wild-type littermates (WT) at 24 h after HI. As shown in Figure 1A,B, CaMKK β KO mice had larger infarcts than the WT mice (WT 40.06 ± 2.50% versus KO 52.61 ± 1.78%, ** *p* < 0.01; *n* = 6–7 per group).

### 2.2. Inhibition of CaMKK Produced Larger Infarct Volume in Wild-Type (WT) Mice at 24 h Post-HI

A pan inhibitor, STO-609, was administered intraperitoneally to WT mice to confirm the detrimental effects of CaMKK genetic deletion. Inhibition of CaMKK β exacerbated HI-induced infarct size (Figure 2A,B, vehicle treated 48.25 ± 2.10% versus STO-609 treated 55.12 ± 2.25%, * *p* < 0.05; *n* = 10 per group). Our findings indicated that the genetic deletion and pharmacological inhibition of CaMKK β aggravated the brain injury after HI in WT mice. It is noteworthy that the control groups of Figure 1 and Figure 2 did not have similar infarct sizes, which highlights the importance of using littermates as appropriate controls for KO mice experiments.

### 2.3. STO-609 Suppressed Functional Recovery at 3 Weeks after HI 

Long-term disability is a major consequence of HI. Therefore, STO-609 was used to examine the roles of CaMKK β during chronic survival. A multitude of behavioral assessments were performed in neonatal mice 21 days after HI. There was no significant difference in seizure scores between the two groups (Figure 3A). However, in the wire hanging test, vehicle-treated mice exhibited a significantly longer latency to fall from the wire than STO-609 treated mice (Figure 3B vehicle-treated 31.35 ± 1.96s versus STO-609-treated 24.11 ± 2.28s, * *p* < 0.05; *n* = 7 per group). Conversely, in the corner test, the percentage of right turns in STO-609-treated mice was significantly higher than in the vehicle-treated mice (Figure 3C vehicle-treated 51.43 ± 4.85% versus STO-609-treated 71.43 ± 6.61%, * *p* < 0.05; *n* = 7 per group). However, no significant difference was seen in seizure scores (Figure 3A). Overall, we found that the inhibition of CaMKK β significantly worsened behavioral recovery after HI.

### 2.4. Inhibition of CaMKK Increased HI-Induced Tissue Loss after Long-Term Survival 

Using CV staining, we found that the inhibition of CaMKK β by STO-609 increased the amount of brain tissue loss 21 days after HI (Figure 4A,B vehicle-treated 33.39 ± 3.44% versus STO-609-treated 50.71 ± 6.96%, * *p* < 0.05; *n* = 6 per group). Together, the results of the histological analysis and functional tests suggested that CaMKK β is neuroprotective in neonatal mice after HI insult, even after long-term survival.

### 2.5. Inhibition of CaMKK Reduced Levels of pCaMK IV, Bcl-2, Collagen IV and Claudin-5 and Increased Levels of Bax at 6 h after HI

Next, we explored the molecular mechanisms of CaMKK β during acute survival after HI in mice using STO-609, an inhibitor of CaMKK β. The data showed that after HI, the inhibition of CaMKK β reduced levels of pCaMK IV, one of the major downstream targets of CaMKK (Figure 5A,B). Bcl-2 is an anti-apoptotic protein and its transcription is suggested to be regulated by CaMKK signaling [7]. Our data demonstrated that STO-609 downregulated the levels of Bcl-2 and increased levels of Bax, a pro-apoptotic protein (Figure 5C–F), suggesting the role of CaMKK in anti-apoptosis in HI. We also observed a reduction in the levels of collagen IV and claudin-5, two important components of the BBB with CaMKK β inhibition after HI (Figure 6A–D), implying exacerbated BBB impairment.

## 3. Discussion

This study reported several novel findings. First, we found that the inhibition of CaMKK β, either pharmacologically or genetically, exacerbated the infarct size of the neonatal brain after a hypoxia–ischemia insult. Second, in chronic survival experiments, the inhibition of CaMKK β by STO-609 worsened functional recovery and increased brain tissue loss. Third, we investigated the effect of CaMKK signaling in BBB integrity and apoptosis in neonates after HI. We showed that the pharmacological inhibition of CaMKK β reduced the levels of anti-apoptotic protein Bcl-2 and increased pro-apoptotic molecule Bax. The reduction of key BBB proteins (collagen IV and claudin-5) was observed after CaMKK inhibition in HI. Our study has identified CaMKK as a novel biological target for HI treatment.

Our present study is the first to report that CaMKK is neuroprotective in neonatal hypoxia–ischemia brain injury. CaMKK is a major kinase activated by intracellular calcium with two major isoforms. It resides in both the cytosol and the nucleus, where it responds to changes in Ca^2+^ levels and propagates the CaMK cascade signal. CaMKK is held in an inactive state by its auto-inhibitory domain, which interacts with the catalytic domain to prevent kinase activity. The binding of Ca^2+/^CaM releases this auto-inhibitory domain, thus activating the kinase. CaMKK then activates its two primary downstream targets, CaMK I and CaMK IV, through phosphorylation [17]. In a developing brain, the role of CaMKK has been well documented with a focus on axonal growth, dendritic arborization, and the formation of dendritic spines and synapses [17]. However, little is known regarding CaMKK in neurological disorders, particularly ischemia, in neonates. The exacerbated HI-induced outcome in CaMKK β KO mice and in STO-609-treated mice suggests that this molecule normally mediates neuroprotective signaling. 

Our data showed that CaMKK regulates apoptosis in the neonatal stroke brain, as pharmacological inhibition of CaMKK β after HI downregulated the level of Bcl-2 and upregulated the pro-apoptotic protein, Bax. This is consistent with our previous work showing that CaMKK β KO brains display lower Bcl-2 compared with WT controls in adult mice after stroke [18], further supporting the role of CaMKK in cell survival post-stroke. The mechanism of CaMKK regulating Bcl2 is likely through transcriptional activation, which is the result of CaMK IV activation, histone acetylation and chromatin conformational changes [11]. CaMK IV is a known neuroprotective molecule in this setting of cerebral ischemia. The activation of CaMK IV signaling pathways protects against neuronal injury in the hippocampal CA1 subfield after transient global ischemia [19]. Moreover, the overexpression of CaMK IV reduced mouse cortical neuronal injury after oxygen glucose deprivation in vitro [20]. In a focal stroke model in adult mice, our previous study has shown that CaMK IV deletion worsened outcomes [7]. In our current study, STO-609, a CaMKK inhibitor, also reduced the phosphorylation of CaMK IV 6 h after stroke onset, indicating that CaMK IV mediates the effect of CaMKK. Bcl2 is a neuronal survival factor and its expression is regulated by histone deacetylase 4 (HDAC4) [11]. We previously showed that the deletion of CaMKK β or CaMK IV in adult mice increased HDAC4 nuclear translocation and the gene-repressing function of HDAC4 [13,14]. Therefore, the effect of CaMKK seen in the current study may be through the CaMK IV/HDAC4/Bcl2 pathway. However, their molecular interactions in a neonatal brain after stroke warrants future investigation. 

It has been previously suggested that CaMKK may protect the BBB integrity by reducing Matrix metalloproteinases(MMPs), the cleaving enzyme for important components of the BBB such as collagens and basal lamina [21,22]. CaMKK inhibition enhanced MMP2 and MMP9 activity while reducing levels of key BBB proteins in adult stroke models [7]. Here we showed that in a neonatal hypoxia–ischemia model, the pharmacological inhibition of CaMKK also decreased collagen IV and laminin protein levels. The mechanisms through which CaMKK prevents BBB protein breakdown are unknown, but may involve MMPs as previously suggested. In this study, we injected STO-609 systemically, therefore it is possible that this influences neutrophils, which are a source of MMPs. Additionally, we have shown that endothelial cells also express CaMKK [16], the inhibition of which has been shown to exacerbate endothelial cell survival by activating the CaMK IV/SIRT pathway after a stroke. Thus, STO-609 may also reduce brain endothelial CaMKK to directly regulate BBB protein expression. Our current and previous studies suggest that the CaMKK pathway plays a beneficial role in BBB integrity after cerebral ischemia across the life span. 

There were some limitations in this study. First, changes in the CaMK IV and other molecules after pharmacological treatment may be epiphenomenal and not directly mechanistic. We did not use any drugs (CaMK IV inhibitor) or any CaMK IV KO mice to clarify the specific role of CaMK IV in the neonatal model. CaMKK has another downstream target, CaMK I, which may contribute to the effect of CaMKK in this model as well. Second, we only assessed BBB integrity and the apoptosis pathway, however, other pathologies of stroke may be influenced by CaMKK signaling. For example, we have previously shown that CaMKK is a major suppressor of inflammatory responses in adult stroke models. Third, we only used males in this study. However, the role of CaMKK may be different between male and females as HI injury has been shown to be sexually dimorphic [23]. Our group has previously shown that males and females display different infarcts after HIE [23]. The underlying mechanism may involve different immune responses in microglia in different sexes. CaMKK indeed plays a critical role in inflammation and its effects in neonatal stroke may differ between the sexes. Therefore, we did not inter-mix males and females in any individual experiment performed in the study to avoid the potential difficulties of data interpretation. The main focus of the study was to define the functional role of CaMKK in neonates after stroke using pharmacological and genetic approaches. We will further probe the roles of this kinase in females in the near future to improve our understanding of CaMKK in neontatal stroke. More detailed mechanisms will be revealed with future studies. 

In summary, we established the neuroprotective role of CaMKK in HI using a neonatal mice model. Hopefully pharmaceutical companies will develop small activators that can specifically enhance the activity of this kinase. The timing of neonatal stroke occurrence is often unknown due to the delay in clinical presentation. Therefore, treatment may be delayed. In the future, once a specific CaMKK activator is available, we will test its efficacy and therapeutic window in a HIE model. It is interesting that this pathway inhibits apoptosis, which takes place in a delayed manner after injury. Our data additionally suggested CaMKK reduces long-term behavior deficits. We believe CaMKK activation may be particularly helpful in neonatal stroke in which treatment is often delayed and patients suffer long-term disability. 

## 4. Materials and Methods

### 4.1. Animals and Treatments

CaMKK β KO mice were a gift from Dr. Anthony Means at Duke University and were backcrossed into a C57BL/6J background for over 9 generations. The wild-type C57BL/6 mice were obtained from the Jackson Laboratory. We used male P10 pups (KO and their littermates) for HI surgery. All procedures were performed in accordance with NIH guidelines for the care and use of laboratory animals and were approved by the Institutional Animal Care and Use committee of the University of Texas Health Science Center. STO-609, a pan CaMKK inhibitor (30 μM, 50 μL, dissolved in 0.1% DMSO) was administered intraperitoneally in male WT mice immediately after hypoxia-ischemia. The same amount of 0.1% DMSO or STO-609 was used in the sham group. All analyses and surgeries were performed by the researchers who were blinded to the treatment (drug and mouse phenotype). 

### 4.2. Animals Model of Hypoxic–Ischemic Encephalopathy (HIE)

Mouse HI was performed according to a method described previously [1,24]. Briefly, postnatal day (P10) C57BL/6 male mice were anesthetized with isoflurane (4% for induction and 1.5–2% for maintenance). A midline cervical incision was made, and the right common carotid artery was exposed and double ligated with 6-0 silk suture thread. The wound was then sutured with 4-0 surgical silk. Surgery time for each pup did not exceed 5 min. After recovering with their dams for 2 h, mice were put in a chamber containing 10% oxygen and 90% nitrogen for 40 min. Next, the animals were placed on a temperature-controlled blanket for 20 min and then returned to their dams.

### 4.3. Infarct Volume Qualification

Briefly, the brains of the mice were removed at 24 h after HI and sectioned into 1 mm slices, then immersed into 2% TTC solution at 37 °C for 10 min, followed by 4% paraformaldehyde. The infarct volume was traced and analyzed by Sigma-scan Pro5 software (https://systatsoftware.com/products/sigmascan/) [1]. 

### 4.4. Neurobehavioral Tests

Seizure score: Behavioral features characteristic of seizures were assessed for a duration of 5 min at 24 h after HI, the animals were assigned a score for the highest level of seizure activity observed during that period, as follows: 0 = normal behavior; 1 = immobility; 2 = rigid posture; 3 = repetitive scratching, circling, or head bobbing; 4 = forelimb clonus, rearing, and falling; 5 = mice that exhibited level four behaviors repeatedly; and 6 = severe tonic–clonic behavior [25,26].

Wire hang test: The wire hang test was performed at 21 days after HI. A wire cage top (18 inch × 9 inch) with its edges taped off was used for this experiment. The mice were placed on the metallic wire, the top was inverted and latency to fall was recorded. This was repeated three times with an interval of 5 min between attempts. The cutoff time was set at 60 s. Mean latency to fall was determined [27].

Corner test: The corner test was evaluated at 21 days after HI. A corner was made by attaching two boards (30 cm × 20 cm × 1 cm) at an angle of 30°. A small opening was made along the joint to encourage entry into the corner. Animals were placed midway from the corner. When the animals reached the corner and their vibrissae were stimulated by the walls, the animals with cerebral ischemia neglected the damaged side, and reared to the intact side (the right side) when the cardboard stimulated the vibrissae. The number and direction of rears were recorded for 20 trials, and the percentage of right turns was calculated. Only turns involving full rearing along either board were recorded. This test was used to assess the sensorimotor and postural function [28].

### 4.5. Brain Atrophy Measurement

At 3 weeks (P31) after HI, the animals were anesthetized with tribromoethanol (Avertin^®^ intraperitoneal injection at a dose of 0.25 mg/g body weight). Animals were perfused transcardially with ice cold 0.1 M sodium phosphate buffer (pH 7.4) followed by 4% paraformaldehyde (PFA); the brain was removed from the skull and post-fixed in 4% PFA over night and subsequently placed in 30% sucrose. The brain was cut into 30-μm free-floating coronal sections on a freezing microtome and every eighth slice was stained with cresyl violet ([CV] Sigma, St. Louis, MO, USA) for evaluation of damage [29].

### 4.6. Western Blotting

Western blotting was performed as described previously [18]. Briefly, at 6 h after HI, mouse brains were harvested (*n* = 3 in each group) and proteins of the ipsilateral hemisphere were extracted by homogenizing in RIPA lysis buffer (Boston Bioproducts, Ashland, MA, USA ) containing phosphatase inhibitor (04906845001, Roche, Indianapolis, IN, USA) and protease inhibitor (04693116001, Roche, USA) and the protein concentration was determined with a BCA assay (Thermo Scientific, Waltham, MA, USA). Equal amounts of protein (50 μg) were loaded on an SDS-PAGE gel. After being electrophoresed and transferred to a polyvinylidene fluoride membrane, the membrane was blocked by immersion for 1 h in Tris-buffered saline (TBS) containing 5% dry milk and incubated with the primary antibody overnight at 4 °C. The primary antibodies used were rabbit polyclonal (anti-phospho T196 + T200 CaMKIV antibody, 1:1000, ab59424, Abcam; anti-Collagen IV antibody, 1:500, ab-19808, Abcam; anti-Claudin-5 antibody, 1:500, ab-15106, Abcam; anti-Bax antibody, 1:1000, #2772, Cell Signaling Technology; or anti-BCL-2 antibody, 1:1000, #3498, Cell Signaling Technology, (Danvers, MA, USA) diluted in Tris-buffered saline containing 0.1% Tween-20 and 4% bovine serum albumin or 5% fat free milk. After incubation, the membranes were washed at least three times with TBST (TBS containing 0.2% Tween-20) and were then incubated again for 1 h with the appropriate secondary antibodies (anti-rabbit IgG, 1:5000, Cell Signaling Technology; or anti-mouse IgG, 1:5000, Vector, (Danvers, MA, USA), which were diluted in the same blocking buffer at room temperature. The membranes were washed with TBST three times. Finally, an electrochemiluminescence detection kit (Thermo Scientific) was used for signal detection. β-actin (primary antibody 1:5000; Sigma) was used as a loading control. The data were analyzed by the software Quantity one 4.6.1 (Bio-Rad, Hercules, CA, USA).

### 4.7. Data Analysis

All the data were expressed as mean ± SEM. Statistical differences between more than two groups were analyzed by using one-way ANOVA followed by Tukey post-hoc analysis. Statistical difference between two groups was analyzed using a t-test. *p* value < 0.05 was considered statistically significant.

## Figures and Tables

**Figure 1 ijms-20-02063-f001:**
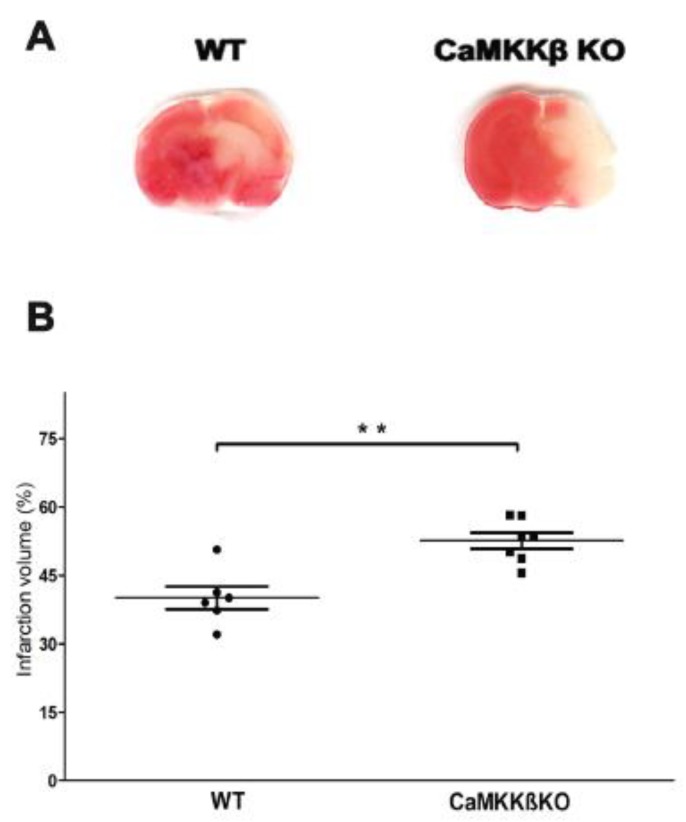
Calcium/calmodulin-dependent protein kinase kinase (CaMKK) β knockout (KO) mice had larger infarcts than the wild-type (WT) mice, assessed 24 h after HI. (**A**) Representative TTC stained coronal brain sections from WT and CaMKK β KO HI mice are shown. (**B**) Quantification of infarct volume revealed that deletion of CaMKK β produced elevation in the infarct volume (*n* = 6–7 per group, ** *p* < 0.01 versus WT HI; vertical bars indicate SEM).

**Figure 2 ijms-20-02063-f002:**
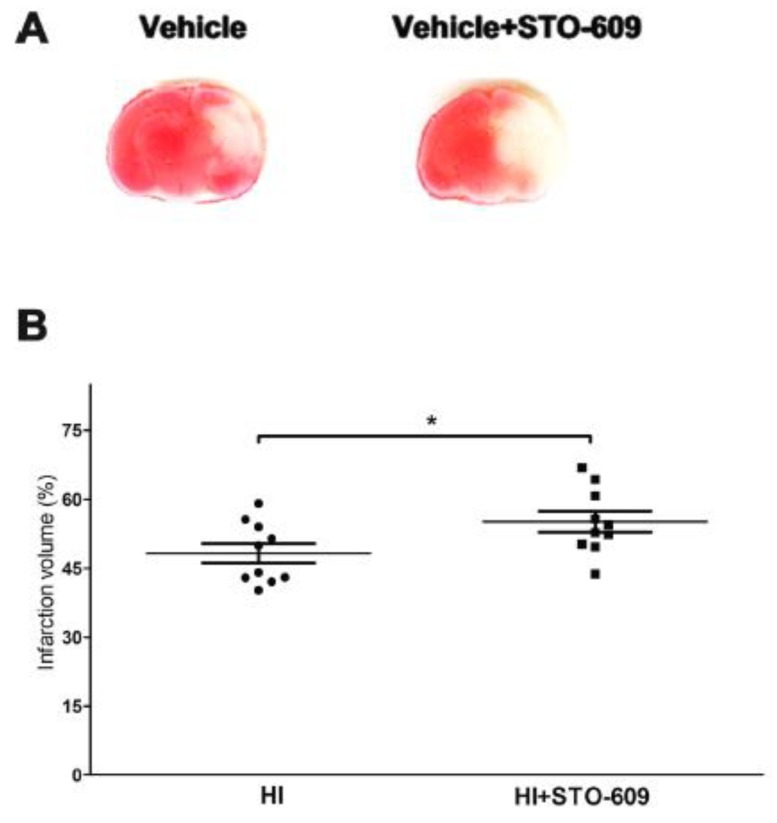
STO-609 increased infarct volume in wild-type (WT) mice at 24 h after HI. (**A**) Representative TTC stained coronal brain sections from vehicle and STO-609 treated HI mice are shown. (**B**) Quantification of infarct volume revealed that inhibition of CaMKK β increased infarct volume (*n* = 10 per group, * *p* < 0.05 versus vehicle treated mice; vertical bars indicate SEM).

**Figure 3 ijms-20-02063-f003:**
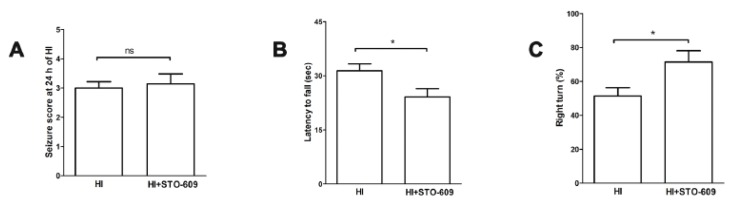
The inhibition of CaMKK β suppressed functional recovery at 3 weeks after HI. (**A**) Seizure scores were not significantly different between vehicle- and STO-609-treated groups at 24 h after HI. (**B**) and (**C**) Functional outcomes were evaluated by the wire hanging test and corner test at 3 weeks post-HI. The percentage of latency to fall and right turns were significantly different between the vehicle- and STO-609-treated groups (*n* = 7 per group, * *p* < 0.05 versus vehicle-treated mice; vertical bars indicate SEM). ns: No significate.

**Figure 4 ijms-20-02063-f004:**
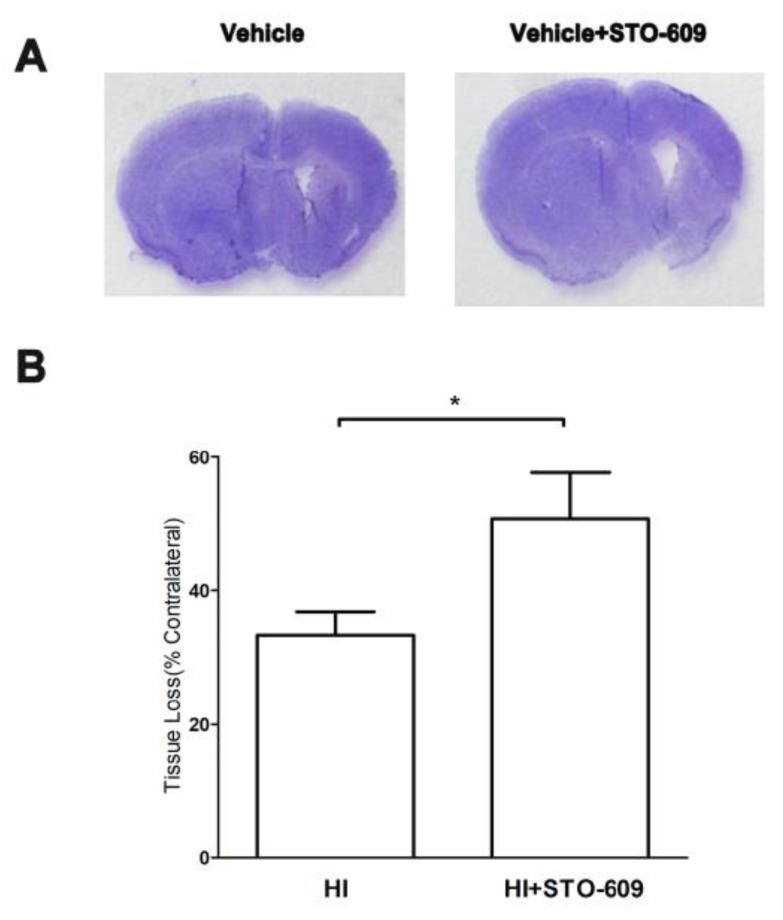
The suppression of CaMKK β increased HI-induced tissue loss after long-term survival. (**A**) Representative brains with tissue loss in vehicle- and STO-609-treated mice 21 days after HI. (**B**) Quantification of brain tissue loss resulting from HI, (*n* = 6 for each group, * *p* < 0.05 versus vehicle treated mice; vertical bars indicate SEM).

**Figure 5 ijms-20-02063-f005:**
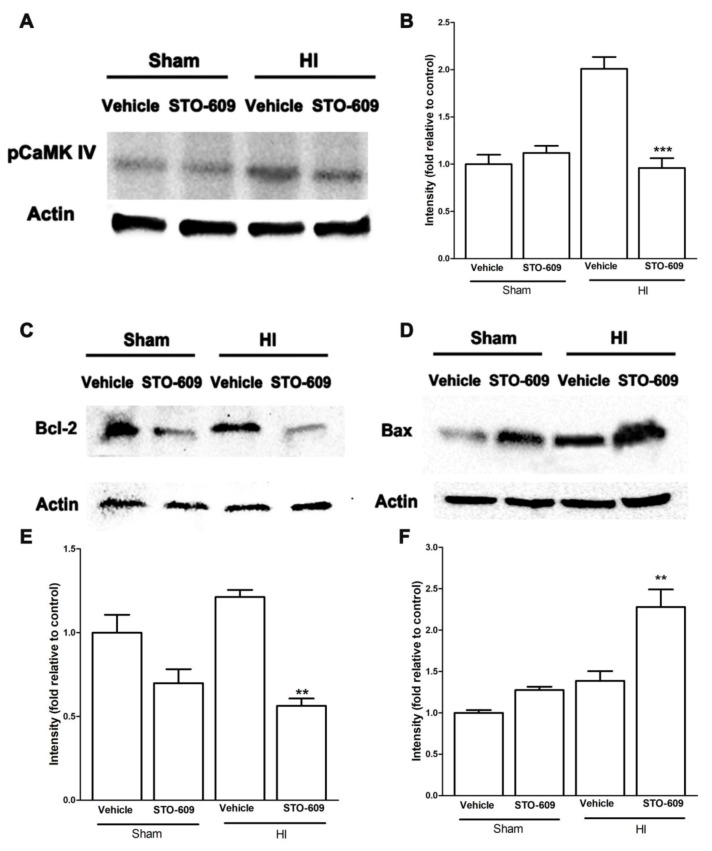
The inhibition of CaMKK β reduced pCaMK IV, Bcl-2, collagen IV and claudin-5 but increased Bax after HI. (**A**,**C**,**D**) Representative western blots of pCaMK IV, Bcl-2 and Bax in vehicle and inhibitor groups. (**B**,**E**,**F**) Quantification of pCaMK IV, Bcl-2 and Bax expression, (*n* = 3 for each group, ** *p* < 0.01, *** *p* < 0.001 versus vehicle treated HI mice; vertical bars indicate SEM).

**Figure 6 ijms-20-02063-f006:**
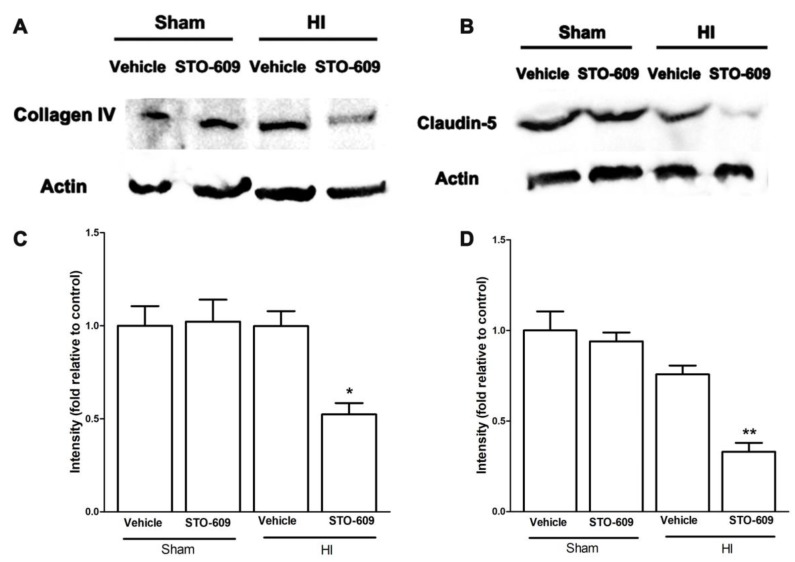
Representative western blots of collagen IV (**A**) and claudin-5 (**B**) in vehicle and inhibitor groups. (**C**,**D**) Quantification of collagen IV and Claudin-5 expression, (*n* = 3 for each group, * *p* < 0.05, ** *p* < 0.01 versus vehicle treated HI mice; vertical bars indicate SEM).

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
