# Peer review of "Inhibition of Calcium/Calmodulin-Dependent Protein Kinase Kinase β Is Detrimental in Hypoxia–Ischemia Neonatal Brain Injury"

_ijms, 2019, doi:10.3390/ijms20092063_

Reviewer 1 Report

1. In figure 3, the graphs are presented in a different order than what is presented in the text. It would make more sense to have fig 3A be first in the text and on the graphs. 

2. Figure 5 - the text is too small to read. Either break into two figures to make each item larger, or make each item larger. Currently cannot read the legends on the graphs making this figure useless. 

3. Only using male gender is a very large limitation; the authors are strongly encouraged to use female offspring as well to strengthen the findings of this study. 

4. It is suggested by the authors that the CaMKK pathway is a potential mechanism for treatment. However, when stroke or HIE occurs in infants, the timing is often unknown. How can this pathway be utilized in a situation that would be clinically relevant? IE - is there a way to enhance this pathway that can be done in a timing that would correlate to the clinical situation?

Author Response

1. Thanks for your wonderful suggestion. We have rearranged the figures as suggested. There was no significant difference in seizure scores between two groups (Figure 3 A). However, in the wire hanging test, vehicle-treated mice exhibited a significantly longer latency to fall from the wire than STO-609 treated mice (Figure 3 B vehicle-treated 31.35±1.96s versus STO-609 treated 24.11±2.28s, *P < 0.05; n=7 per group).  We also highlighted the changes in the manuscript.

2. Thanks for your kind suggestion. In the revised manuscript, Figure 5 has been broken into two figures (5&6) and we have made all panels larger. The figure legends have also been reorganized. We highlighted the changes in the manuscript.
3. We totally agree with this reviewer on this point. We are aware that stroke is sexually dimorphic disease in neonates. Our group has previously shown that males and females display different infarcts after HIE (Mirza et al.. 2015). The underlying mechanism may involve different immune responses in microglia in different sexes. CaMKK indeed plays critical role in inflammation and its effects in neonatal stroke may differ in sexes. Therefore, we did not inter-mix males and females in any individual experiment performed in the study to avoid the potential difficulties of data interpretation. We have added the limitation of using males only into our discussion. We will further probe for role of this kinase in females in the near future to improve our understanding of CaMKK in neontatal stroke. Page 9.

4. This is a great point. Our study has identified CaMKK as a potential target for neonatal stroke therapy. Hopefully pharmaceutical companies will develop small activators that can specifically enhance the activity of this kinase. The timing of neonatal stroke occurrence is often unknown due to the delay in clinical presentations. Therefore treatment may be delayed. In the future, once a specific CaMKK activator is available, we will test its efficacy and therapeutical window in a HIE model. It is interesting this pathway inhibits apoptosis, which takes place in a delayed manner after injury. Our data additionally suggested CaMKK reduces long-term behaviour deficits. We believe CaMKK activation may be particularly helpful in neonatal stroke in which treatment may be often delayed and patients suffer long-term disability. We have added the clarifications in the discussion. Pages 9.

Reviewer 2 Report

Thank you for the opportunity to review this interesting manuscript. This manuscript describes a potential neuroprotective role for CaMKKbeta in neonatal HI, and demonstrates that CaMKK inhibition via the pharmacological agent, STO-609, increases infarct volume, and was associated with decreased Bcl-2, collagen IV and claudin-5 and increased Bax expression.  I have some comments that would improve the quality of this manuscript:

1. CaMKK also phosphorylates AMPK (mention of this should be included in the introduction, as phosphorylation of AMPK has been shown to be neuroprotective in neonatal HI mouse models).

2. There is a large variation in infarction volume between figures 1 and 2. For Figure 1, the WT mice had an infarct size of 40.6% +/- 2.5%, whereas the vehicle treated WT mice in Figure 2 had an infarct size of 48.25 +/-2.1% (which is very similar to the size observed in the CaMKK KO mice [52.61% +/- 1.78%]). Can the authors please provide comment on this variation?

3. For section 2.4 (line 100), suppression is not the correct way to describe the action of STO-609, instead 'Inhibition' of CaMKK 'activity' would be a better description for this section.

4. For Figure 5 - please include the changes in expression for total CaMKIV, and describe as proportion of CaMKIV phosphorylated.

5. For Figure 5, please clarify how the data was normalised. Some appear to be normalised to vehicle sham, e.g. Bcl-2, pCaMKIV, Collagen IV, however others appear to be normalised to vehicle HI, e.g. claudin-5.  All should be normalised in the same manner.

6. Was AMPK phosphorylation measured following STO-609 treatment?

7. Were changes in phospho CaMKIV and Bax observed in the CaMKK KO mice?

There are some methodological factors that require clarification:

8. Why was saline used as a vehicle control when STO-609 was dissolved in DMSO?

9. For section 4.3 (Infarct volume qualification) - were animals perfused prior to brain removal?

10. How were the brains homogenised for western blot?

11. Were whole brains lysed, or only specific regions?

12. it is stated that the lysis buffer contains 'phosphorylation inhibitors' - please clarify this statement

13. For the western blot, was 50mg of protein added to each lane? This is a very high amount of protein

14. The CaMKIV antibody described in section 4.6 should be corrected to phospho T196 + T200 CaMKIV, rather than just CaMKIV.

Author Response

Thank you for reviewing and your suggestions were be very helpful for us to improve the quality of this manuscript.

1. Thanks for your wonderful suggestion. This information is very important to this work and we state it in the introduction.

“…AMP-activated protein kinase (AMPK), a sensor of cellular stress, may also be activated by CaMKK during the depletion of ATP or alterations in intracellular calcium concentrations [12, 13]. It has been shown that inhibition of AMPK prior to HI results in a worse outcome in mouse models of HI [14]….” The changes were highlighted in the manuscript. Page 1

2. This is a wonderful question. The Figure 1, the WT used were C57 black mice purchased from a vendor while in Figure 2, control WT mice were actually littermates of CaMKK KO. It has been suggested crossing a targeted gene to a control mouse background does not necessarily clean the original genetic background (Holmdahl & Malissen 2012). The difference between the controls in two figures here actually highlights the importance of using littermates for KO as the appropriate controls. We have added this into our result description. Page 3.
3. Thanks for the correction. In the revised manuscript, we have changed to 2.4. Inhibition of CaMKK increased HI-induced tissue loss after long-term survival.Page 4

4. Thanks for your helpful advice. We hypothesize that CaMKK phosphorylates its downstream kinases including CaMK IV. We did not anticipate any changes of the total CaMK IV by the treatment of STO-609. Therefore actin was used as loading controls. We have added this clarification into the revised manuscript. Page 6.
5. We apologized for the inconsistency. Claudin-5 has been normalised to vehicle sham in the revised manuscript. Page 7

6. We have previously shown that STO-609 treatment did not alter AMPK phosphorylation in the cerebral ischemic setting (Li et al., Stroke 2013), indicating these two molecules may not interact in stroke. Therefore we focused on BBB and apoptosis pathways in the current study. We have added this in the discussion. Page 9.
7. We did not measure these molecule in the CaMKK beta KO model.  To investigate the role of CaMKK in BBB impairment and apoptosis after neonatal stroke, we used a pharmacological inhibitor. We agree if we had measured these molecules in the beta isoform KO mice, we would have provided more isoform specific information as our inhibitor is a pan inhibitor. We have made sure that no overstatement in the isoform specificity when CaMKK inhibitor was used was made throughout the revised manuscript.

8. We apologize for the error. 0.1% DMSO was used as a vehicle control. We have made changes accordingly in the revised manuscript. Page 9
9. Thanks for your question. For CV staining, Animals were perfused transcardially with ice cold 0.1 M sodium phosphate buffer (pH 7.4), however, the mice were not perfused to brain removal for TTC staining. We corrected it in the section 4.3 and highlighted it. Page 10.

10. The brains were firstly homogenised in the homogenizer on the ice by hands then homogenised by a sonicator on the ice (5 seconds one time for three times; amplitude was 60%). We have provided these technical details in the revised manuscript. Page 10.
11. Thank you. The ipsilateral hemispheres were used for WB. We have clarified in the revised manuscript “Proteins of the ipsilateral hemisphere were extracted by homogenizing in…” Page 11.

12. Thanks for your question. We have clarified this statement in the revised manuscript and highlighted it. “…phosphatase inhibitor (04906845001, Roche, Indianapolis, IN,USA) and protease inhibitor (04693116001, Roche,USA) ….”. Page 11.

13. Thanks for the great point. We had to use a very high amount of protein as phosphosphorylated CaMKIV was very difficult to detect. We have added that clarification in the methods. Page 11.
14. Thanks for your question. The correction has been made in the revised manuscript and highlighted. “anti-phospho T196 + T200 CaMKIV antibody, …”. Page 11.

Round  2

Reviewer 1 Report

The authors have responded to all points well and the manuscript is improved.